
# Decoupling impacts of weather conditions on interannual variations in concentrations of criteria air pollutants in south China – constraining analysis uncertainties by using multiple analysis tools

Yu Lin[1], LeimingZhang[2*], Qinchu Fan[3], He Meng[4], Yang Gao[1,3,5], Huiwang Gao[3,5], Xiaohong Yao[1,3,5*]

[1]Sanya Oceanographic Institution (Ocean University of China), Yazhou Bay Science & Technology City, Sanya, China
[2]Air Quality Research Division, Science and Technology Branch, Environment and Climate Change Canada, Toronto, ON, M3H 5T4, Canada
[3]Key Laboratory of Marine Environment and Ecology (MoE), Ocean University of China, Qingdao, China
[4]Qingdao Eco-Environment Monitoring Center of Shandong Province, Qingdao, China
[5]Laboratory for Marine Ecology and Environmental Sciences, Qingdao National Laboratory for Marine Science and Technology, Qingdao, China

*Correspondence to*: Leiming Zhang (leiming.zhang@ec.gc.ca), Xiaohong Yao (xhyao@ouc.edu.cn)

**Abstract.** In this study, three methods including the random forest (RF) algorithm, boosted regression trees (BRTs) and the improved complete ensemble empirical mode decomposition with adaptive noise (ICEEMDAN) were adopted for

investigating emission-driven interannual variations in concentrations of air pollutants including $PM_{2.5}$, $PM_{10}$, $O_3$, $NO_2$, CO, $SO_2$ and ($NO_2+O_3$) monitored in six cities in south China from May 2014 to April 2021. The first two methods were used to calculate the deweathered hourly concentrations, and the third one was used to calculate decomposed hourly residuals. To constrain the uncertainties in the calculated deweathered or decomposed hourly values, a self-developed method was applied to calculate the range of the deweathered percentage changes (DePCs) of air pollutant concentrations in annual scale.

Emission-driven trends and emission-driven percentage changes (PCs) during the whole seven-year period were generated with the four methods being applied to analyzing the data. The consistency in the trends between the RF-deweathered and BRTs-deweathered concentrations and the ICEEMDAN-decomposed residuals of an air pollutant in a city reaches approximately 70% of all the studied cases, but that in the PCs reaches only approximately 30% of all the cases. The remaining cases with inconsistent trends and/or PCs indicated large uncertainties produced by one or more of the three

methods. The calculated PCs from the deweathered concentrations and decomposed residuals were thus combined with the corresponding range of DePCs calculated from the self-developed method to gain the robust range of DePCs where applicable. Building on the robust ranges, the mitigation effects were discussed.

## 1 Introduction

With rapid economic growth in the past several decades across China, air pollution has become increasingly severe in most
parts of the country (Chan and Yao, 2008; He et al., 2002). A turning point emerged in the most recent decade, benefited from stringent emission control measures implemented in China since 2013, such as "Atmospheric Pollution Prevention and



Control Action Plan" (APPCAP) (Chen et al., 2020; Vu et al., 2019; Zhang et al., 2020). Trends in long-term monitored pollutants are important indicators of the effectiveness of the emission control policies (Hogrefe et al., 2000; Rao et al., 1997). This is particularly true in China, where air pollutant emissions have not been updated in the annual reports of ecology and environment issued by local governments at the city level since 2014. The Multi-resolution Emission Inventory for China (MEIC) was developed in 2012 by Tsinghua University to estimate anthropological air pollutant emissions, but it was updated every two-three year and only up to 2017.

To evaluate existing national emission control strategies in China (such as APPCAP), several studies have analyzed air pollutants concentrations measured at the national monitoring stations (Hu et al., 2021; Xu and Zhang, 2020; Zhao et al., 2021). However, trends and interannual variations in air concentrations of the monitored pollutants were affected by not only emission changes but also varying meteorological conditions and/or weather systems (Dang et al., 2021; Lin et al., 2021; Vu et al., 2019; Zhang et al., 2019a; Zhang et al., 2019b; Zhao et al., 2020; Henneman et al., 2015; Foley et al., 2015; Astitha et al., 2017; Hogrefe et al., 2002). For example, Zhao et al. (2020) reported that the observed large declines in $PM_{2.5}$, $SO_2$ and CO concentrations on the national scale during the COVID-19 outbreak were primarily caused by poor dispersion meteorological conditions. Vu et al. (2019) argued that the $PM_{2.5}$ target of 60 $\mu g \cdot m^{-3}$ would have not been achieved in Beijing in the winter of 2017 if without the favourable weather conditions for rapid dispersion and precipitation scavenging of air pollutants. Similarly, Lin et al. (2021) suggested that meteorological factors significantly reduced $O_3$ concentrations from 2013 to 2020 in eastern and central China, as indicated by the reversed $O_3$ trends after removing the major meteorological effects. It is thus essential to decouple the total trends in pollutants concentrations into portions caused by varying meteorological factors and weather conditions and by emission changes so that the mitigation effects can be evaluated accurately.

In literature, the multiple linear regression (MLR) method is considered as the simplest approach to decouple the effects of meteorological factors from changed emissions on the trends in air pollutants concentrations (Borlaza et al., 2022; Chen et al., 2020; Li et al., 2019a; Otero et al., 2018; Zhai et al., 2019). However, the MLR analysis sometimes suffers from the auto-correlation inherently existed between different meteorological parameters (Yao et al., 2009). To overcome this weakness, "meteorological normalization" tools have been developed based on statistical modelling (Chen et al., 2020; Gong et al., 2021; Grange and Carslaw, 2019; Li et al., 2020; Xiao et al., 2021; Xue et al., 2020; Zhai et al., 2019). For example, the machine learning techniques, such as the random forest (RF) algorithm and boosted regression trees (BRTs), performed better than the traditional methods like MLR (Carslaw and Taylor, 2009; Grange et al., 2018) or other air quality numerical models like Weather Research and Forecasting-Community Multi-scale Air Quality model (Vu et al., 2019; Foley et al., 2015; Astitha et al., 2017) in analyzing air quality trends and meteorological impacts. These methods have been used widely in relevant studies across China, e.g., in Beijing (Vu et al., 2019), Beijing-Tianjin-Hebei region (Qu et al., 2020) and North China Plain (He et al., 2021b). In the methods mentioned above, meteorological data are a necessity. In contrast, another exiting method called the empirical mode decomposition (EMD) and its updated version the improved complete ensemble empirical mode decomposition with adaptive noise (ICEEMDAN) directly decompose time series of air pollutants



concentrations and deduct the perturbation from meteorological factors on the residuals (trend) to some extent (Colominas et al., 2014; Fu et al., 2020). It should be pointed out that, due to the non-linearity of chemical reactions related to air pollutants, none of the existing methods is perfect in decoupling the effects of dominant factors in the total trends of pollutants concentrations. To evaluate the uncertainties in trend analysis, combining results from several different methods are recommended (Xiao et al., 2021; Xue et al., 2020; Hogrefe et al., 2002).

The updated global air quality guidelines from World Health Origination (WHO) declared in 2021 brought new challenges to policy makers for establishing more stringent emission control policies, even in the relatively clean regions like south China. For example, air quality in Hainan province needs to be further improved to meet the new WHO standards, and the demonstration of Hainan Free Trade Port and declaration of the province as a National Ecological Civilization Demonstration Zone in China make this task more challenging. Even more challenges exist for the cities in Guangdong province because of their higher air pollutant concentrations than in Hainan (Gong et al., 2021; He et al., 2017; Li et al., 2019b; Zhang et al., 2019b). To accommodate these challenges, the effect of APPCAP needs to be first assessed regionally in south China. For this purpose, we analyzed seven-year (May 2014 to April 2021) concentration data of six criteria air pollutants ($PM_{2.5}$, $PM_{10}$, $O_3$, $NO_2$, CO and $SO_2$) as well as the sum of $NO_2$ and $O_3$ in six cities in south China, of which two (Haikou and Sanya) are in Hainan province and four (Guangzhou, Shenzhen, Zhuhai and Zhanjiang) are in Guangdong province. Three different analysis methods were used to identify emission-driven interannual variations and perturbations from varying weather conditions. In addition, a self-developed method was further introduced to constrain analysis uncertainties.

## 2 Materials and methods

### 2.1 Monitoring stations and monitored air pollutant concentrations and meteorological data

Six cities in south China, including Haikou and Sanya in Hainan Province and Guangzhou, Shenzhen, Zhuhai and Zhanjiang in Guangdong province, were selected in the present study (Fig. 1). There are two monitoring stations in Sanya, four in Zhuhai, five in Haikou, six in Zhanjiang and 11 stations in both Guangzhou and Shenzhen (Fig. 1 and Table S1). The hourly average air quality data for six criteria pollutants ($PM_{2.5}$, $PM_{10}$, $O_3$, $NO_2$, CO and $SO_2$) were downloaded from the China National Environmental Monitoring Centre (CNEMC, http://www.cnemc.cn/sssj/) for every monitoring station. The city-specific hourly air quality data represent the average of all the stations in the same city. Data before May 2014 were incomplete, and thus, only the data after May 2014 were used for the analysis. One whole year data covered from May to the next April, e.g., the first-year annual average (referred to as 2014 annual average in the discussion below) covered from May 2014 to April 2015, and the last year average (referred to as annual average in 2020 below) covered from May 2020 to April 2021. For a pollutant in a city, 52204–58695 hourly data were available in seven years (Table S2). Note that all concentrations were converted to the values under the standard conditions (273.15K, 1 atm) for consistency. Note that





concentrations of volatile organic carbons (VOCs) were not reported by CNEMC and this group of pollutants is not considered in the present study.

Hourly meteorological data including wind speed (ws), wind direction (wd), air temperature (air_temp), relative humidity
(rh), precipitation (prep) and dew point (dp) were obtained from the meteorological observational station at a nearby airport (Fig. 1 and Table S1), which are accessible from the NOAA Integrated Surface Database (ISD) by using the "worldmet" R package (Carslaw, 2021). The meteorological data of each city were combined with city-specific hourly air quality data as input for the machine learning analysis.

**2.2 Data analysis methods**

Two machine learning methods, including the RF algorithm and the BRTs, were separately used to calculate the deweathered hourly concentrations. The third method, the ICEEMDAN, was used to decompose hourly residuals of air pollutants. The Mann-Kendall (M-K) method was then applied to the deweathered and decomposed values to extract the trends and calculate the percentage changes (PCs). A self-developed method was further applied to calculate the range of the deweathered percentage changes (DePCs) of air pollutant concentrations in annual scale. The three PCs and DePCs were
combined for constraining the uncertainties and generating a robust range of DePCs. Fig. 2 shows the framework of this study with the four methods to be applied.

The RF algorithm was performed based on the "rmweather" R package (Grange et al., 2018) and the "ranger" R package (Wright and Ziegler, 2017), and the BRTs was performed based on the "deweather" R package (Carslaw et al., 2012; Carslaw and Taylor, 2009). The application of these packages has been well documented in literature, e.g., analyzing long-
115 term trends in concentrations of air pollutants (Grange and Carslaw, 2019; Ma et al., 2021; Mallet, 2020), assessing impact of clean air actions (Vu et al., 2019; Zhang et al., 2020), and evaluating the response of air quality during the COVID-19 lockdown (Dai et al., 2021; Shi and Brasseur, 2020; Wang et al., 2020; Munir et al., 2021; Lovric et al., 2021). The independent input variables included temporal variables (hour, day, weekday, week and month), observational concentrations and meteorological parameters (ws, wd, air_temp, rh, prep and dp). The inputs were divided into two groups:
the training dataset that account for 80% of the data and a testing dataset that contained the remaining 20%. The performance was evaluated by statistical metrics included the correlation coefficient ($R^2$), root mean square error (RMSE), mean bias (MB), mean fractional bias (MFB) and mean fractional error (MFE). The formulas used to calculate RMSE, MB, MFB and MFE were shown as below:

$$\text{RMSE} = \sqrt{\frac{\sum_{i=1}^{N}(P_i-O_i)^2}{N}} \qquad \text{(Equation-1)}$$

$$\text{MB} = \frac{1}{N}\sum_{i=1}^{N}(P_i - O_i) \qquad \text{(Equation-2)}$$

$$\text{MFB} = \frac{2}{N}\sum_{i=1}^{N}\left(\frac{P_i-O_i}{P_i+O_i}\right) \times 100\% \qquad \text{(Equation-3)}$$

$$\text{MFE} = \frac{2}{N}\sum_{i=1}^{N}\left(\frac{|P_i-O_i|}{P_i+O_i}\right) \times 100\% \qquad \text{(Equation-4)}$$



in which $P_i$ and $O_i$ represent the $i$th predicated and observed values, N represents the number of data used to test. Note that United States Environmental Protection Agency (USEPA) proposed the criteria and goal values for MFE and MFB to evaluate the air quality modelling performance, which are MFE≤75% and MFB≤±60% for criteria and MFE≤50% and MFB≤±30% for goal (USEPA, 2007). No such criteria value has been set for the other parameters defined in the above equations.

The indices of PM$_{2.5}$ in Guangzhou are shown as an example in Fig. 3, and the summary of all air pollutants in the six cities can be found in Table S3. In Fig. 3, the minimum RMSE values obtained for the PM$_{2.5}$ test in Guangzhou and used for the final calculation are 20.08 (RF algorithm) and 19.52 (BRTs), respectively. $R^2$ values are 0.80 (RF algorithm) and 0.81 (BRTs), respectively, implying that the predicted values of PM$_{2.5}$ by the two methods only moderately well reproduce the observations. The MB values are 1.18 (RF algorithm) and 0.11 (BRTs), respectively, implying that the BRTs better reproduced the observations than RF algorithm in this case. Note that MB of zero would indicates an ideal prediction. The calculated MB being deviated from zero implies the deweathered hourly concentrations suffering from the errors to some extent, and the errors would automatically transfer into the deweathered trends and PCs. MFB and MFE values were less than 30% and 50%, respectively, for both RF algorithm and BRTs (Fig. 3), satisfying the goal values set by USEPA. This suggests well performance in reproducing observations using the two methods, as listed in Table S3.

The meteorologically normalized air pollutant concentrations at a particular time were calculated by averaging 1000 predictions from the models with meteorological variables randomly resampled from the study period (2014−2020).

$$y_{dew} = \frac{1}{1000}\sum_{i=1}^{1000} x_{i,pred} \qquad \text{(Equation-5)}$$

where $x_{i,pred}$ is the model-predicted concentration for a given meteorological condition at time $i$, and y$_{dew}$ is the corresponding deweathered hourly concentration at a particular time under averaged meteorological conditions. In this study, the deweathered hourly concentrations by the BRTs contained more spikes than those by the RF method. Some of the spikes may be caused by irregular emissions such as agriculture biomass burning, wild forest fires, holiday fireworks, construction activities (Video S1) and accidents (Chen et al., 2017; Dai et al., 2021; Enayati et al., 2021; Chen et al., 2021; Shen et al., 2022), etc.

The ICEEMDAN method (Colominas et al., 2014), which is an improved version of the empirical mode decomposition (EMD) method, overcomes the "end-effect" originally existed in EMD, providing modes with less noise and avoiding the spurious modes. The original data can be decomposed and expressed as:

$$x = \sum_{i=1}^{k} d_i + r \qquad \text{(Equation 6)}$$

where $x$ is the original data, $d_i$ is the $i$th intrinsic mode function (IMF), $k$ is the total number of IMFs, and $r$ is the final residual. This method has been applied in various fields, such as financial prediction (Zhou and Chen, 2021) and air quality assessment (Luo et al., 2020). The implementation of the ICEEMDAN method is based on a Python package named PyEMD (Laszuk, 2017). The number of modes needs to be pre-set in this method, which was chosen based on sensitivity test results with the following two criteria: 1) only one oscillation cycle should be kept in the real residual; and 2) combining the real





residual and the final mode would end up two or more oscillation cycles. For example, the decomposed residual plus the last mode was finally used as the real ICEEMDAN-decomposed residual for PM$_{2.5}$ in Guangzhou, Shenzhen, Zhanjiang and Zhuhai (Fig. S1a-d, f) while the decomposed residual was used directly as the real ICEEMDAN-decomposed residual for PM$_{2.5}$ in Haikou (Fig. S1e). Note that the ICEEMDAN method requires a complete time series of data. Approximately 5%
data were missing for each air pollutant in each city (Table S2), but the missing data did not occur at the identical hour in two consecutive days, except PM$_{10}$ concentration in Zhanjiang. For the special cases of PM$_{10}$ in Zhanjiang, the missing data were replaced by the average values of the observed data between the nearest day before and after at the identical hour. This approach of replacing missing data may introduce a small uncertainty on the decomposed residuals.

In this study, we also developed a novel method to study emission-driven interannual variations in air pollutant
concentrations by calculating the range of DePC on an annual scale based on an earlier approach proposed by Yao and Zhang (2020) (referring to the self-developed method in the present study). Details of this method are presented in the Supplement, with an example of calculating the range of DePC of PM$_{2.5}$ concentration between two years (May 2020 – April 2021 relative to May 2014 – April 2015) in Guangzhou (Table S4 and Fig. S2). There are five steps in this method, including 1) reconstructing the time series of data in any two years to the same size; 2) conducting correlation analysis using the
reconstructed data in any two years and removing outliers after the inflection point (Fig. S2); 3) repeating step 2) to remove more outliers; 4) calculating the range of DePC; and 5) evaluating residual perturbations by varying weather conditions. The main advantages of this method include 1) avoiding the calculation of the deweathered hourly concentrations or decomposed hourly residuals of air pollutants in which their uncertainties are unpredictable; 2) confirming the accuracy of DePC when the range of DePC is sufficiently narrow; and 3) identifying the large perturbation from varying weather conditions on DePC
when the range of DePC is broad.

The M-K analysis is employed to resolve the trends in the time series of the annual average concentration of each pollutant. Qualitative trend results revolved by the M-K method include 1) an increasing/decreasing trend with a P value of <0.05; 2) a probable increasing/decreasing trend with a P value of 0.05–0.1; 3) a stable trend with a P value of >0.1 as well as with a ratio of <1.0 between the standard deviation and the mean of the dataset; and 4) a no-trend for P>0.1 with all the other
conditions (Aziz et al., 2003; Kampata et al., 2008; Yao and Zhang, 2020).

## 3 Results

### 3.1 Trends and PCs of PM$_{2.5}$ and PM$_{10}$

The seven-year (2014–2020) average mass concentrations of PM$_{2.5}$ were the highest in Guangzhou at 34 μg·m$^{-3}$, followed by 27 μg·m$^{-3}$ in Shenzhen, 26 μg·m$^{-3}$ in Zhanjiang and Zhuhai, 20 μg·m$^{-3}$ in Haikou and 15 μg·m$^{-3}$ in Sanya (Table 1). The
annual average PM$_{2.5}$ concentrations in most cities and in nearly all the years (Table S5) exceeded the annual average Class-I level (15 μg·m$^{-3}$) of Ambient Air Quality Standards (AAQS) in China, and exceeded the latest WHO air quality guideline values by several times.



The largest decreases in the annual average $PM_{2.5}$ mass concentration from the first year (2014) to the last year (2020) occurred in Guangzhou, i.e., by 17 $\mu g \cdot m^{-3}$ (or 39%) (Table 1 and Fig. 4). A significant decreasing trend was also identified during the seven-year period by the M-K method ($p<0.05$). A similar case was also found in Shenzhen with a decrease of 9 $\mu g \cdot m^{-3}$ (or 28%) from 2014 to 2020 and a significant decreasing trend ($p<0.05$) during the same period. However, a probable decreasing trend ($0.05 \leq p < 0.1$) or a stable trend ($p \geq 0.1$) was revealed by the M-K method in the other four cities. Note that a 20%–40% decrease in $PM_{2.5}$ annual average concentrations was frequently observed across China since 2013, e.g., a nationwide decrease by an overall 22% from 2015 to 2018 (Zhao et al., 2021), an approximate 40% decrease in Beijing and 20% decrease in Pearl River Delta from 2015 to 2019 (Hu et al., 2021; Xu and Zhang, 2020).

To explore the emission-driven trends in $PM_{2.5}$ concentration in the six cities, the RF-deweathered and BRTs-deweathered $PM_{2.5}$ concentrations and the ICEEMDAN-decomposed residuals of $PM_{2.5}$ concentrations are examined in Fig. 4a-l. In Guangzhou and Shenzhen, a consistent decreasing trend ($p<0.05$) was identified by the M-K method in the deweathered $PM_{2.5}$ concentrations and the decomposed residuals of $PM_{2.5}$ concentrations (Table 1 and Fig. 4a and b). The PCs from 2014 to 2020 were also reasonably consistent between the different datasets mentioned above (Table 2), i.e., with the standard deviation of the three PCs being within 10% of the corresponding mean absolute value. Specifically, the PCs from 2014 to 2020 in Shenzhen calculated from the RF-deweathered and BRTs-deweathered $PM_{2.5}$ concentrations and the ICEEMDAN-decomposed residuals were -29%, -32% and -30%, respectively, which were not much different from that using the original $PM_{2.5}$ concentration (28% as discussed above). A combination of these four PCs values in Shenzhen allowed to infer that: 1) the reduced air pollutant emissions in Shenzhen and upwind regions likely decreased the $PM_{2.5}$ concentrations by 30% ± 1.5% (mean ± standard deviation) from 2014 to 2020, and 2) the perturbation from varying weather conditions cancelled out 2% ± 1.5% out of the of 30% ± 1.5% decrease. In Guangzhou, the PCs of $PM_{2.5}$ concentrations from 2014 to 2020 estimated by the three methods were -33% (RF-deweathered), -35% (BRTs-deweathered) and -35% (ICEEMDAN-decomposed), while the PCs calculated from the original annual average $PM_{2.5}$ concentrations was -39%, as mentioned above. Thus, the reduced emissions of air pollutants in Guangzhou and upwind regions likely decreased the concentrations of $PM_{2.5}$ by 34% ± 1% during the seven-year period, while the perturbation from varying weather conditions caused an additional decrease of 5% ± 1%. Gong et al. (2021) also reported an additional 5% decrease driven by varying meteorological conditions, on top of the 47% decrease driven by reduced emissions, in the national annual averages of $PM_{2.5}$ mass concentration from 2013 to 2019 in China.

A decreasing trend ($p<0.05$) was also identified in Zhuhai, Haikou and Sanya when using the RF-deweathered and BRTs-deweathered concentrations and the ICEEMDAN-decomposed residuals (Table 1 and Fig. 4d-e, j-l), which are in contrast with probably decreasing trends generated from using the original $PM_{2.5}$ concentration data. The perturbations from varying weather conditions on $PM_{2.5}$ mass concentrations likely complicated the effects of reduced air pollutant emissions in the three cities and upwind regions during 2014–2020. It is noted that the PCs estimated from the three different methods (RF-deweathered, BRTs-deweathered and ICEEMDAN-decomposed) varied little for Sanya (-23%, -21% and -24%) and Haikou





(-19%, -20% and -20%) from 2014 to 2020, but quite large for Zhuhai (-35%, -36% and -26%), the latter case was likely due to the large uncertainties associated with one or more methods (Table 2).

A no-trend or stable trend was identified for Zhanjiang (Table 1, Fig. 4c and i), regardless of which method was used. The PCs from 2014 to 2020 were all positive, i.e., 14% (RF-deweathered), 3% (BRTs-deweathered), 5% (ICEEMDAN-decomposed), and 8% (original data), indicating emission-driven increases in $PM_{2.5}$ concentration in this city during this period.

Similar analysis to the one discussed above was also conducted to $PM_{10}$ concentrations (Tables 1 and 2 and Fig. S3a-l), results from which can be summarized below.

1) The highest seven-year (2014–2020) average $PM_{10}$ concentrations of 57 μg·m$^{-3}$ occurred in Guangzhou, followed by 45 μg·m$^{-3}$ in Shenzhen, 43 μg·m$^{-3}$ in Zhuhai, 42 μg·m$^{-3}$ in Zhanjiang, 37 μg·m$^{-3}$ in Haikou and 29 μg·m$^{-3}$ in Sanya. The annual average $PM_{10}$ concentrations exceeded the annual average Class-I level (40 μg·m$^{-3}$) of AAQS in China in most cities and most years, and exceeded the latest WHO air quality guideline values by 2–4 times.

2) The M-K analyses showed either a no or stable trend during 2014–2020 if using the original annual average $PM_{10}$ concentrations in the six cities (Table 1). Inconsistent trends were then obtained between using the three different methods (RF-deweathered, BRTs-deweathered and ICEEMDAN-decomposed) in five out of the six cities. The only exception is for Guangzhou in which a decreasing trend was identified from all of the three methods, although a no trend was extracted from the original annual average concentrations. For Shenzhen, a decreasing trend was obtained using the RF-deweathered method while a no or stable trend was obtained from the BRTs-deweathered and ICEEMDAN-decomposed method. For Sanya, a decreasing trend was obtained using the RF-deweathered and the ICEEMDAN-decomposed method, while a no-trend was obtained using the BRTs-deweathered method. The inconsistency between the trends extracted by the three different methods was mostly because the actual interannual changes, and thus the magnitudes of the trends, were small, which are on the same order of magnitude to the methodology uncertainties. Combining all the trends generated using the three different methods and the original data, we concluded a slightly decreasing or stable trend in emission-driven $PM_{10}$ concentrations for all the cities.

3) The PCs of $PM_{10}$ concentration from 2014 to 2020 in Guangzhou were consistent between using the three different methods e.g., -15% (RF-deweathered), -13% (BRTs-deweathered) and -15% (ICEEMDAN-decomposed), while that from using the original $PM_{10}$ concentration data, -11%. Thus, reduced emissions of air pollutants in Guangzhou and upwind regions likely decreased $PM_{10}$ concentrations by 14% ± 1% during the seven-year period, while the perturbation from varying weather conditions cancelled out 3% ± 1%. The reasonably consistent PCs were also obtained for Shenzhen and Zhanjiang, although with inconsistent decreasing trends. However, inconsistent PCs were obtained from the three different methods for the other three cities due to methodology uncertainties and the actual small trends, as explained above.

**3.2 Trends and PCs of $O_3$ and ($NO_2$+$O_3$)**





Among the four gaseous criteria pollutants, $O_3$ concentrations in the six cities exceeded the most, on percentage basis, the
Class-I levels of AAQS in China (Table S5). Trend analyses were conducted for both $O_3$ and ($NO_2$+$O_3$) considering the
titration reaction of $O_3$ with NO to form $NO_2$ in ambient air (Chan and Yao, 2008; Li et al., 2019a; Seinfeld and Pandis, 1998;
Sicard et al., 2020; Wang et al., 2017).

The seven-year (2014–2020) average concentrations of $O_3$ were highest at 69 μg·m$^{-3}$ in Zhanjiang and Zhuhai, followed by
62 μg·m$^{-3}$ in Shenzhen, 60 μg·m$^{-3}$ in Haikou, 58 μg·m$^{-3}$ in Sanya (Table 1), and the lowest at 51 μg·m$^{-3}$ in Guangzhou. The
titration reaction of $O_3$ with NO likely decreased $O_3$ concentrations to some extent in Guangzhou, as implied by the highest
annual average $NO_2$ concentrations in this city among the six cities (Table 1). In contrast, the highest $O_3$ annual averages
occurred in Zhanjiang and Zhuhai. The annual average $NO_2$ concentrations in the two cities were smaller than that in
Guangzhou, but larger than that in Sanya. Thus, both the reduced depletion of $O_3$ via the titration reaction and the enhanced
photochemical formation of $O_3$ likely contributed to the highest $O_3$ annul averages in the two cities (He et al., 2021a; Liu et
al., 2021; Shen et al., 2021).

Using the original data of annual average $O_3$ concentrations (Table 1 and Fig. 5a-f), the M-K analysis results showed an
increasing trend in Zhanjiang, Shenzhen, Haikou and Guangzhou (p<0.05), and a no- trend in Zhuhai and Sanya. Using the
RF-deweathered concentrations, the BRTs-deweathered concentrations and the ICEEMDAN-decomposed residuals (Fig. 5a-
l and Fig. S1g-l), M-K analysis results generated the same trend as mentioned above in every city. Thus, the emission-driven
increasing trends in $O_3$ concentration from 2014 to 2020 can be firmly confirmed in four cities (Zhanjiang, Shenzhen,
Haikou and Guangzhou).

The PCs of the deweathered concentrations, the decomposed residuals, and the original annual average concentrations from
2014 to 2020 in the four cities with increasing trends of $O_3$ concentration were further analyzed, and were presented below
from the largest to the smallest PCs. In Haikou, the PCs from 2014 to 2020 was 65% based on the original annual average $O_3$
concentrations (Table 2). The corresponding PCs were 54%, 65% and 64% based on the RF-deweathered concentrations, the
BRTs-deweathered concentrations, and the ICEEMDAN-decomposed residuals, respectively. Combining these numbers
together, we concluded that the emission changes of $O_3$ precursors likely increased the $O_3$ concentration by at least 54% from
2014 to 2020, and the perturbations from varying weather conditions seemingly yielded an additional increase of 0%−11%.
Similarly, in Guangzhou, Shenzhen and Zhanjiang, the emission changes of $O_3$ precursors likely increased the concentrations
of $O_3$ by 26%±1.5%, >10% and >17%, respectively, from 2014 to 2020, and the perturbations from varying weather
conditions seemingly yielded an additional increase of 14%±1.5%, 8%–18%, and -1%–14%, respectively.

In the case of ($NO_2$+$O_3$), an increasing trend (p<0.05, Table 2) was obtained from 2014 to 2020 in Haikou, while probable
increasing, no-trend or stable trends were obtained in the other five cities based on the original annual average concentration
data (Fig. S4a-l). A consistent increasing trend in ($NO_2$+$O_3$) was obtained in Guangzhou and Zhanjiang based on any of the
RF-deweathered concentrations, BRTs-deweathered concentrations, and decomposed residuals of ($NO_2$+$O_3$). In Haikou, an
increasing trend was obtained based on the RF-deweathered and BRTs-deweathered concentrations while a probable
increasing trend was obtained from the decomposed residuals. The increasing trends in ($NO_2$+$O_3$) from 2014 to 2020 in the





above-mentioned three cities confirmed the enhanced formation of $O_3$. However, either no or stable trends were obtained in Zhuhai, Shenzhen and Sanya based on the deweathered concentrations or the decomposed residuals of $(NO_2+O_3)$ (Table 1).

The contrasting trends between $(NO_2+O_3)$ and $O_3$ in Shenzhen, i.e., a no-trend in the former and an increasing trend in the latter (Table 1), was likely due to the reduced $O_3$ depletion via the titration reaction of $O_3$ by NO.

The PCs in $(NO_2+O_3)$ from 2014 to 2020 in Haikou, Guangzhou and Zhanjiang were presented below from the largest to the smallest PCs. In Haikou, the PCs were estimated to be 45%, 55%, 48% and 62% based on the RF-deweathered concentrations, the BRTs-deweathered concentrations, the ICEEMDAN-decomposed residual and the original annual

average concentrations, respectively. The first three PCs values for $(NO_2+O_3)$ were smaller than those of $O_3$ by 9%−16%. Thus, the reduced $O_3$ depletion via the titration reaction likely increased the $O_3$ concentrations by 9%–16%, leaving the major portions of $O_3$ increases (45%–55%) from 2014 to 2020 attributed to the emission-driven enhanced $O_3$ formation (Li et al., 2019a; Wang et al., 2017). In Guangzhou, the estimated four PCs in $(NO_2+O_3)$ were 11% (RF-deweathered), 7% (BRTs-deweathered), 15% (ICEEMDAN-decomposed) and 15% (original data). These numbers were smaller than those for

$O_3$ by 11%–25%, implying similar contributions from the reduced $O_3$ depletion via the titration reaction and the enhanced $O_3$ formation to the total increased $O_3$ concentration. In Zhanjiang, the estimated four PCs in $(NO_2+O_3)$ were 18%, 13%, 20% and 14%, which are mostly similar to those for $O_3$ (18%, 17%, 32% and 18%) (Table 2), implying the dominate contribution of the enhanced $O_3$ formation to the increased $O_3$ concentration.

### 3.3 Trends and PCs of $NO_2$, CO and $SO_2$

The seven-year (2014–2020) average $NO_2$ concentrations were the highest at 46 μg·m⁻³ in Guangzhou, followed by 30 μg·m⁻³ in Shenzhen, 29 μg·m⁻³ in Zhuhai, 15 μg·m⁻³ in Zhanjiang and Haikou, and lowest at 12 μg·m⁻³ in Sanya (Table 1). Annual average $NO_2$ concentrations in Guangzhou exceeded the annual average Class-I level of AAQS in China (40 μg·m⁻³) in all the years except 2020, partly due to reduced emissions in 2020 as a result of COVID-19 pandemic (Bauwens et al., 2020; Shi and Brasseur, 2020; Wang et al., 2020; Wang et al., 2021; Zhao et al., 2020). Annual average $NO_2$ concentrations were

below 40 μg·m⁻³ in all the other cities during all the years, but were far above the latest WHO air quality guideline value of 10 μg·m⁻³.

A decreasing trend in $NO_2$ concentration from 2014 to 2020 was obtained in Shenzhen and Zhuhai based on the deweathered concentrations and the decomposed residuals, while a probable decreasing trend was obtained based on the original annual average concentration data (Fig. S5a-l). In Shenzhen, the PCs in $NO_2$ from 2014 to 2020 were mostly consistent between the

different methods (Table 2), e.g., -18% (RF-deweathered), -20% (BRTs-deweathered), -21% (ICEEMDAN-decomposed) and -21% (original data). However, this was not the case in Zhuhai for which the four PCs were -17% (RF-deweathered), -16% (BRTs-deweathered), -8% (ICEEMDAN-decomposed) and -18% (original annual average). A stable trend in $NO_2$ concentration from 2014 to 2020 was obtained in Guangzhou, regardless of the method used. The impact of the reduced $NO_x$ emissions in Guangzhou and/or upwind areas could not be detected in the observed $NO_2$ concentrations on annual scale.





Inconsistent trends were obtained between using different methods in Zhanjiang, Haikou and Sanya, similar to the cases of several other pollutants discussed above.

The annual average concentrations of CO and $SO_2$ were all below the Class-I levels of AAQS in China during 2014–2020 in all the cities (Table 1). A consistent decreasing trend in annual average CO concentration was obtained, regardless which method was used, in all the cities except Haikou (Fig. S6a-l). A consistent decreasing trend in annual average $SO_2$

concentration was also obtained using the four different methods in Shenzhen and Zhuhai. In Guangzhou, a decreasing trend in annual average $SO_2$ concentration was obtained based on the deweathered concentrations and decomposed residuals, while a probable decreasing trend was obtained based on the original annual average data (Fig. S7a-l). In Sanya, an increasing trend in annual average $SO_2$ concentration was obtained based on the deweathered concentrations and decomposed residuals, while a probable increasing trend was obtained based on the original annual average data. In Haikou and Zhanjiang,

inconsistent trends were obtained between using the deweathered concentrations and decomposed residuals.

The reasonably consistent PCs in annual average CO concentration from 2014 to 2020 between using different methods were only obtained in Shenzhen and Zhanjiang, i.e., -40% and -32% (RF-deweathered), -36% and -34% (BRTs-deweathered), -39% and -32% (ICEEMDAN-decomposed), and -36% and -34% (original data), respectively. The PCs in $SO_2$ from 2014 to 2020 were reasonably consistent between using different methods in Guangzhou and Zhuhai, e.g., the four values in Guangzhou

were -46% (RF-deweathered), -46% (BRTs-deweathered), -47% (ICEEMDAN-decomposed) and -44% (original average).

## 4 Discussion

The M-K analysis using the RF-deweathered concentrations, the BRTs-deweathered concentrations and the ICEEMDAN-decomposed residuals during 2014–2020 yielded consistent trends in approximately 70% of the cases. The remaining 30% inconsistent trends were apparently caused by methodology uncertainties in some or all of the three methods (RF, BRTs and

ICEEMDAN). The PCs from 2014 to 2020 using the same three data sets, although mostly comparable, were only absolutely consistent in approximately 30% of the cases. Thus, the PCs calculated from the above three methods were further assessed using the range of DePCs using the self-developed method introduced in Section 2.

The PCs of $PM_{2.5}$ from 2014 to 2020 varied from -35% to -33% in Guangzhou and from -29% to -32% in Shenzhen as discussed in Section 3. After applying the self-developed method, the corresponding DePCs were estimated to be in the

range of -37% – -33% in Guangzhou and -36% – -31% in Shenzhen. The overlap portion between the range of PCs and the range of DePCs in each city was thereby set up as the robust range of DePCs, i.e., -35% – -33% in Guangzhou, and -32% – -31% in Shenzhen (Table 2). The robust ranges of DePCs were almost the same as those of PCs in both cities, further confirming the emission-driven PCs and the perturbation from varying weather conditions presented in Section 3. These cases were referred to as Category 1-a below.

The PCs of $PM_{2.5}$ from 2014 to 2020 were from -20% to -19% in Haikou and from -24% to -21% in Sanya. The corresponding DePC had no overlap with these PCs ranges in these cities, implying nonexistence of a robust range of DePC.



These cases were referred to as Category 1-b below. Note that the self-developed method would not introduce additional observational variables in the calculation process, and the true DePC should be within the range of DePC calculated using the self-developed method.

The PCs of $PM_{2.5}$ from 2014 to 2020 were in a relatively large range in Zhuhai and Zhanjiang, implying certain extent of inconsistence between the three methods (RF, BRTs, and ICEEMDAN). The PCs in Zhuhai varied from -36% to -26%, and the corresponding DePC had no overlap with this range, implying non-existence of a robust range of DePC. This case was referred to as Category 2-a below. The PCs in Zhanjiang ranged from 3% to 14%, and the corresponding DePC completely overlapped this range, again confirming the emission-driven PCs and the perturbation from varying weather conditions presented in Section 3. This case was referred to as Category 2-b below.

The PCs of $PM_{10}$ from 2014 to 2020 were inconsistent between using the RF-deweathered concentrations, the BRTs-deweathered concentrations and the ICEEMDAN-decomposed residuals of $PM_{10}$ in Zhuhai and Sanya. Nevertheless, a robust range of DePCs was obtained in Zhuhai (-20% – -14%), Sanya (-27% – -23%). Comparing the robust range of DePCs with the range of PCs calculated using the original annual average data (-21% in Zhuhai and Sanya), the perturbation from varying weather conditions yielded an additional decrease in $PM_{10}$ concentration by 1%–7% in Zhuhai, but offset by 2%–6%

in Sanya. These cases were refereed as Category 2-c below, featuring a narrower robust range of DePCs than that of PCs calculated from the deweathered concentrations and decomposed residuals. $PM_{10}$ in Shenzhen, Zhanjiang and Guangzhou followed into Category 1-a, confirming the emission-driven PCs and the perturbation from varying weather conditions. $PM_{10}$ in Haikou followed into Category 2-a with no robust range of DePC.

$O_3$ in Haikou, Shenzhen and Zhuhai followed into Category 2-c, and that in Guangzhou, Zhanjiang and Sanya followed into Category 1-a, 2-a and 2-b, respectively. Results of ($NO_2+O_3$), $NO_2$, CO and $SO_2$ also followed into some of the five categories (Table 2), and the above interpretation for $PM_{2.5}$, $PM_{10}$ and $O_3$ in each category on the emission-driven PCs and the perturbation from varying weather conditions are also applicable to the ($NO_2+O_3$), $NO_2$, CO and $SO_2$.

## 5 Conclusions

In this study, we first applied separately the RF algorithm, the BRTs algorithm, and the ICEEMDAN to obtain time series of the deweathered concentrations or decomposed residuals of criteria air pollutants and ($NO_2+O_3$) from May 2014 to April 2021 in the six cities in south China. We found that the RF-deweathered and BRTs-deweathered concentrations and the ICEEMDAN-decomposed residuals yielded consistent trends in approximately 70% of the cases. We then calculated the PCs between the first and the last year using the above-mentioned deweathered concentrations and residuals. Only in

approximately 30% of the cases the PCs were reasonably consistent between the three methods, indicating large methodology uncertainties in one or more methods. The self-developed method was further used to calculate the range of DePCs, and a robust range of DePCs was identified in approximately 70% of the cases.



Based on those consistent trends obtained from the different methods and the robust range of DePCs, we finally generated the following findings.

1)    Significant decreasing trends in $PM_{2.5}$ concentration during 2014–2020 were identified in Guangzhou and Shenzhen, which were mainly caused by the reduced air pollutant emissions and to a much less extent by weather perturbations. A stable or no trend in $PM_{2.5}$ was identified in Zhanjiang, implying no detectable effects of the reduced air pollutant emissions on the monitored $PM_{2.5}$. A decreasing or probably decreasing emission-driven trends were obtained in the remaining cities. The emission-driven effects likely took the lead in the overall changes, although uncertainties

associated with one or more methods still existed on basis of inconsistent PCs.

  2)    Increasing trends in $O_3$ concentration during 2014–2020 were identified in Zhanjiang, Shenzhen, Guangzhou and Haikou. The emission changes of $O_3$ precursors played a dominant role than did the perturbations from varying weather conditions. However, increasing trends in $(NO_2+O_3)$ were only identified in Zhanjiang, Guangzhou and Haikou with increasing and probable increasing trends obtained from different methods, which also confirmed the different

contribution ratios of the reduced $O_3$ depletion via the titration reaction and the enhanced formation of $O_3$.

This study demonstrates the necessity of combining multiple decoupling and/or trend analysis methods in order to constrain the uncertainties in trend analysis results inherent in any individual method. Interpretation of trend analysis results presented in this study could be strengthened if detailed discussions on atmospheric processes and chemistry mechanisms were provided, which unfortunately could not be accommodated here due to the lack of reliable long-term data of concerned

chemical species, such as the major chemical components in $PM_{2.5}$ and $PM_{10}$, VOCs, and up-to-date emission inventory of all the involved pollutants.

**Acknowledgement**

This work was supported by Hainan Provincial Natural Science Foundation of China (grant no.422MS098), Natural Science Foundation of China (grant no. 41776086) and Hainan Provincial Postgraduate Innovative Research Project (grant no.

Yhys2021-5).

*Code and data availability.* The code of DePC calculation can be accessed via https://pypi.org/project/DePC/, the data used in this paper are downloadable from http://www.cnemc.cn/sssj/.

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





**Figure 1: Maps of the study areas and locations of air quality monitoring stations (red star) and one meteorological station (blue triangle) in each city.**





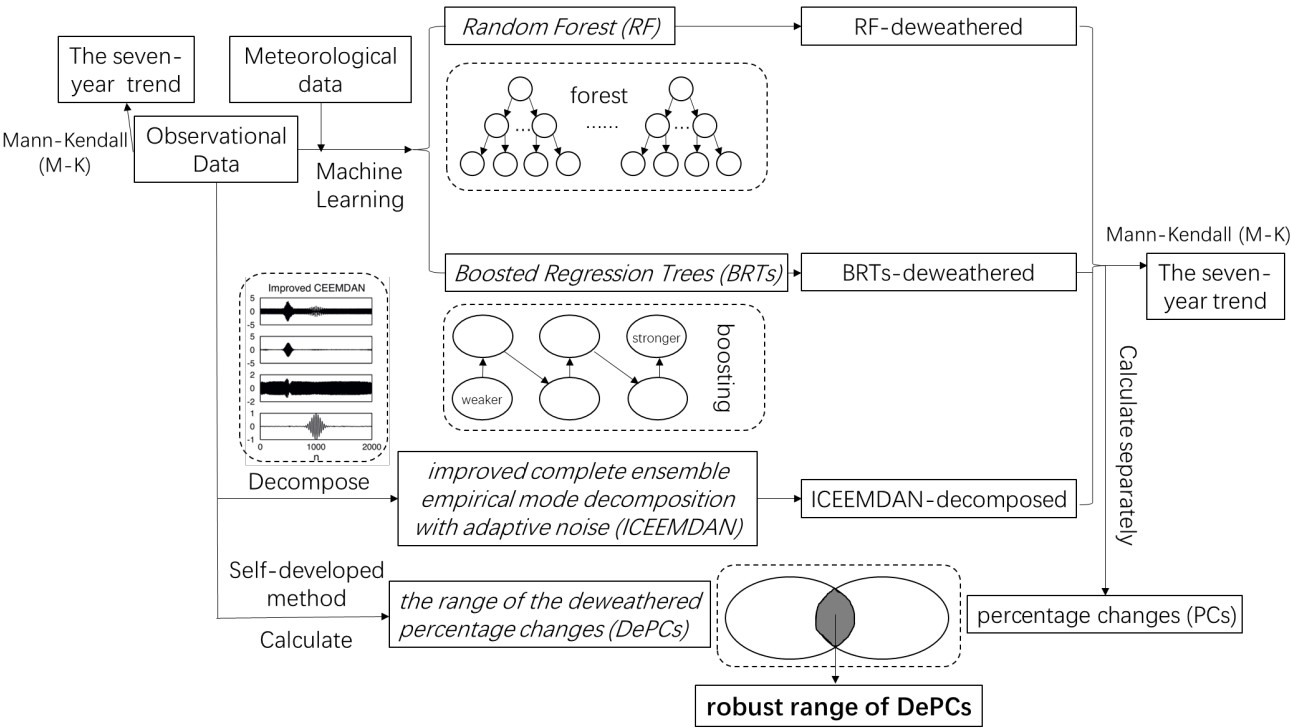

**Figure 2: The framework of this study on the four methods to be applied.**





590

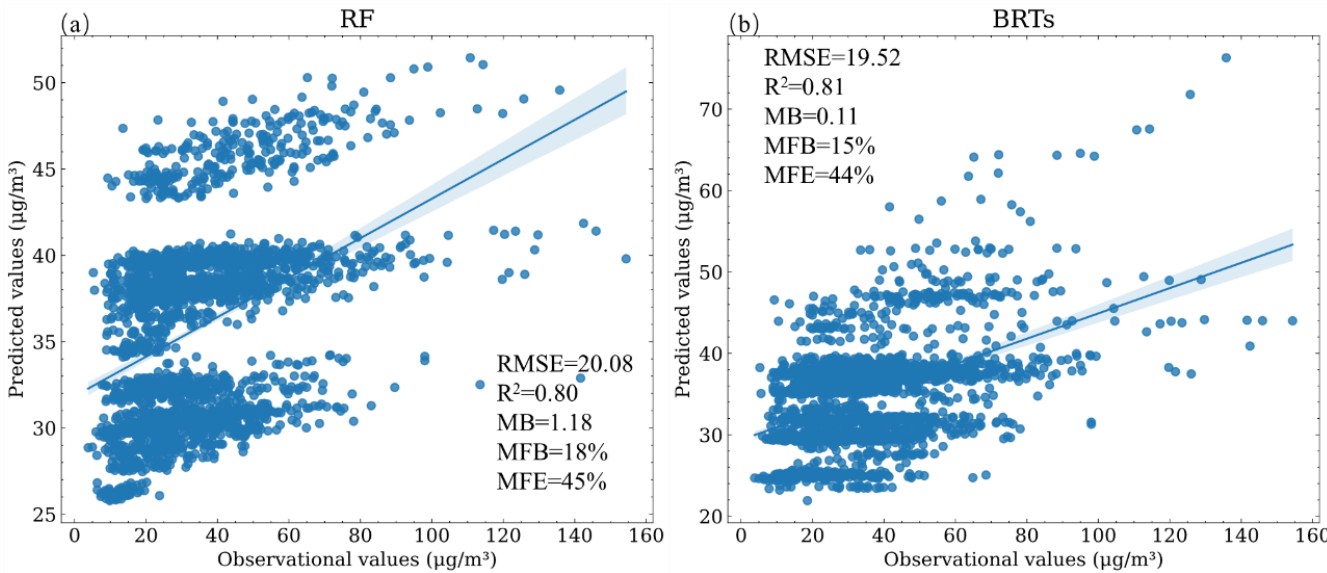

**Figure 3: The performance of PM$_{2.5}$ predictions of the two machine learning methods in Guangzhou. (a) RF-deweathered, (b) BRTs-deweathered.**







**Figure 4: The RF-deweathered and BRTs-deweathered concentrations, ICEEMDAN-decomposed residuals (or mode + residuals of PM₂.₅) and annual averages from May 2014 to April 2021. a-f: deweathered concentrations in the six cities (the order of the cities is same as that listed in Table 1); g-l: decomposed residual or (the last mode + residual) and annual averages in the six cities (* represents the time series of values to be used to calculate the trend and PC).**





**Figure 5: Same as Figure 4, except the pollutant to be O₃.**



**Table 1: The original annual average concentrations of six criteria air pollutants and ($NO_2$+$O_3$) and their trends in original annual averages, RF-deweathered and BRTs-deweathered concentrations and ICEEMDAN-decomposed residuals in six cities detected by the M-K method.**

| Pollutant | city | seven-year average | Annual average | | | | | | | original | RF | BRTs | ICEE-MDAN |
| --- | --- | --- | --- | --- | --- | --- | --- | --- | --- | --- | --- | --- | --- |
| | | | 2014 | 2015 | 2016 | 2017 | 2018 | 2019 | 2020 | trend | | | |
| **$PM_{2.5}$ ($\mu g \cdot m^{-3}$)** | Guangzhou | 34 | 44 | 37 | 37 | 36 | 30 | 28 | 27 | ↓* | ↓* | ↓* | ↓* |
| | Shenzhen | 27 | 32 | 27 | 28 | 28 | 23 | 25 | 23 | ↓* | ↓* | ↓* | ↓* |
| | Zhanjiang | 26 | 24 | 28 | 27 | 28 | 23 | 27 | 26 | — | N | — | N |
| | Zhuhai | 26 | 33 | 27 | 26 | 27 | 24 | 22 | 21 | ↓ | ↓* | ↓* | ↓* |
| | Haikou | 20 | 23 | 21 | 21 | 20 | 16 | 17 | 18 | ↓ | ↓* | ↓* | ↓* |
| | Sanya | 15 | 18 | 15 | 14 | 15 | 14 | 13 | 14 | ↓ | ↓* | ↓* | ↓* |
| **$PM_{10}$ ($\mu g \cdot m^{-3}$)** | Guangzhou | 57 | 63 | 59 | 60 | 57 | 49 | 55 | 56 | N | ↓* | ↓* | ↓* |
| | Shenzhen | 45 | 56 | 44 | 44 | 46 | 39 | 44 | 45 | — | ↓* | N | — |
| | Zhuhai | 43 | 53 | 48 | 43 | 43 | 36 | 37 | 42 | N | ↓* | ↓* | — |
| | Zhanjiang | 42 | 50 | 44 | 43 | 43 | 35 | 40 | 42 | N | — | N | — |
| | Haikou | 37 | 42 | 39 | 39 | 37 | 32 | 34 | 35 | N | ↓* | N | — |
| | Sanya | 29 | 34 | 28 | 28 | 29 | 29 | 29 | 27 | — | ↓* | N | ↓* |
| **$O_3$ ($\mu g \cdot m^{-3}$)** | Zhuhai | 69 | 56 | 66 | 74 | 72 | 67 | 81 | 70 | N | N | N | N |
| | Zhanjiang | 69 | 62 | 60 | 66 | 72 | 71 | 78 | 73 | ↑* | ↑* | ↑* | ↑* |
| | Shenzhen | 62 | 54 | 59 | 63 | 63 | 58 | 69 | 69 | ↑* | ↑* | ↑* | ↑* |
| | Haikou | 60 | 43 | 58 | 61 | 54 | 58 | 76 | 71 | ↑* | ↑* | ↑* | ↑* |
| | Sanya | 58 | 58 | 56 | 57 | 53 | 58 | 64 | 60 | N | N | N | N |
| | Guangzhou | 51 | 45 | 41 | 47 | 51 | 47 | 63 | 63 | ↑* | ↑* | ↑* | ↑* |
| **$NO_2$+$O_3$ ($\mu g \cdot m^{-3}$)** | Zhuhai | 100 | 91 | 100 | 106 | 103 | 98 | 103 | 100 | N | N | N | — |
| | Guangzhou | 99 | 92 | 88 | 101 | 103 | 95 | 110 | 106 | — | ↑* | ↑* | ↑* |
| | Shenzhen | 93 | 89 | 93 | 96 | 93 | 86 | 97 | 97 | N | N | N | N |
| | Zhanjiang | 85 | 79 | 77 | 81 | 88 | 86 | 94 | 90 | ↑ | ↑* | ↑* | ↑* |
| | Haikou | 76 | 58 | 75 | 75 | 68 | 71 | 91 | 94 | ↑* | ↑* | ↑* | ↑ |
| | Sanya | 70 | 72 | 68 | 68 | 66 | 68 | 74 | 72 | N | N | N | N |
| **$NO_2$ ($\mu g \cdot m^{-3}$)** | Guangzhou | 46 | 46 | 45 | 52 | 50 | 46 | 45 | 40 | — | — | — | — |
| | Shenzhen | 30 | 34 | 33 | 32 | 29 | 27 | 27 | 27 | ↓ | ↓* | ↓* | ↓* |
| | Zhuhai | 29 | 34 | 33 | 31 | 29 | 30 | 21 | 28 | ↓ | ↓* | ↓* | ↓* |
| | Zhanjiang | 15 | 16 | 16 | 14 | 15 | 14 | 15 | 16 | — | — | — | N |
| | Haikou | 15 | 15 | 16 | 14 | 13 | 13 | 14 | 18 | N | N | — | ↑* |
| | Sanya | 12 | 14 | 12 | 11 | 12 | 10 | 10 | 12 | — | — | — | ↓* |
| **CO (mg $\cdot m^{-3}$)** | Guangzhou | 0.90 | 0.99 | 0.95 | 0.93 | 0.85 | 0.86 | 0.84 | 0.85 | ↓* | ↓* | ↓* | ↓* |
| | Zhanjiang | 0.77 | 1.01 | 0.87 | 0.79 | 0.72 | 0.68 | 0.66 | 0.67 | ↓* | ↓* | ↓* | ↓* |
| | Shenzhen | 0.76 | 1.04 | 0.82 | 0.80 | 0.68 | 0.64 | 0.65 | 0.67 | ↓* | ↓* | ↓* | ↓* |
| | Zhuhai | 0.66 | 0.81 | 0.68 | 0.71 | 0.61 | 0.63 | 0.58 | 0.60 | ↓* | ↓* | ↓* | ↓* |
| | Haikou | 0.63 | 0.73 | 0.66 | 0.62 | 0.61 | 0.60 | 0.65 | 0.58 | ↓ | ↓ | — | ↓* |
| | Sanya | 0.52 | 0.56 | 0.61 | 0.52 | 0.51 | 0.49 | 0.49 | 0.44 | ↓* | ↓* | ↓* | ↓* |
| **$SO_2$ ($\mu g \cdot m^{-3}$)** | Guangzhou | 11 | 15 | 11 | 12 | 12 | 7 | 8 | 9 | ↓ | ↓* | ↓* | ↓* |
| | Zhanjiang | 8 | 10 | 7 | 9 | 8 | 6 | 9 | 9 | — | — | — | ↓* |
| | Shenzhen | 7 | 9 | 8 | 8 | 8 | 6 | 6 | 7 | ↓* | ↓* | ↓* | ↓* |
| | Zhuhai | 7 | 8 | 10 | 9 | 8 | 5 | 4 | 6 | ↓* | ↓* | ↓* | ↓* |
| | Haikou | 6 | 6 | 6 | 6 | 6 | 5 | 5 | 5 | ↓ | ↓* | — | — |
| | Sanya | 4 | 3 | 4 | 3 | 3 | 4 | 4 | 4 | ↑ | ↑* | ↑* | ↑* |

↑* (↓*): Increasing (Decreasing) trend, i.e., p < 0.05;
↑ (↓): Probably increasing (decreasing) trend, i.e., 0.05 ≤ p < 0.1;
—: Stable trend;
N: No trend.

605




**Table 2: The PCs of six criteria pollutants and (NO₂+O₃) calculated from original averages, RF-deweathered and BRTs-deweathered concentrations and ICEEMDAN-decomposed residuals and the robust ranges of DePC in six cities (units in %, \* represents no robust DePC).**

| Pollutant | city | original | RF | BRTs | ICEE-MDAN | DePC range | final range |
|---|---|---|---|---|---|---|---|
| $PM_{2.5}$ | Guangzhou | -39 | -33 | -35 | -35 | [-37, -33] | [-35, -33] |
| | Zhuhai | -36 | -35 | -36 | -26 | [-41, -39] | \* |
| | Shenzhen | -28 | -29 | -32 | -30 | [-36, -31] | [-32, -31] |
| | Haikou | -22 | -19 | -20 | -20 | [-27, -26] | \* |
| | Sanya | -22 | -23 | -21 | -24 | [-39, -30] | \* |
| | Zhanjiang | 8 | 14 | 3 | 5 | [5,13] | [5, 13] |
| $PM_{10}$ | Zhuhai | -21 | -27 | -24 | -9 | [-20, -14] | [-20, -14] |
| | Sanya | -21 | -19 | -21 | -28 | [-27, -23] | [-27, -23] |
| | Shenzhen | -20 | -23 | -22 | -21 | [-28, -15] | [-23, -21] |
| | Haikou | -17 | -13 | -9 | -13 | [-21, -20] | \* |
| | Zhanjiang | -16 | -19 | -20 | -22 | [-22, -21] | [-22, -21] |
| | Guangzhou | -11 | -15 | -13 | -15 | [-18, -11] | [-15, -13] |
| $O_3$ | Haikou | 65 | 54 | 65 | 64 | [43,59] | [54, 59] |
| | Guangzhou | 40 | 28 | 25 | 26 | [19,34] | [25, 28] |
| | Shenzhen | 28 | 19 | 20 | 10 | [20, 26] | [20, 20] |
| | Zhuhai | 25 | 16 | 10 | 17 | [14, 15] | [14, 15] |
| | Zhanjiang | 18 | 18 | 17 | 32 | [-16, 5] | \* |
| | Sanya | 3 | 3 | 1 | 7 | [0, 4] | [1, 4] |
| $NO_2+O_3$ | Haikou | 62 | 45 | 55 | 48 | [49, 65] | [49, 55] |
| | Guangzhou | 15 | 11 | 7 | 15 | [5,10] | [7, 10] |
| | Zhanjiang | 14 | 18 | 13 | 20 | [-15, 5] | \* |
| | Zhuhai | 10 | 6 | 3 | -1 | [1,23] | [1, 6] |
| | Shenzhen | 9 | 3 | 3 | 3 | [3,16] | [3, 3] |
| | Sanya | 0 | -1 | -1 | 0 | [-3, -1] | [-1, -1] |
| $NO_2$ | Shenzhen | -21 | -18 | -20 | -21 | [-22, -22] | \* |
| | Zhuhai | -18 | -17 | -16 | -8 | [-26,0] | [-17, -8] |
| | Sanya | -14 | -10 | -14 | -9 | [-19, -4] | [-14, -9] |
| | Guangzhou | -13 | -4 | -11 | -13 | [-13, -12] | [-13, -13] |
| | Zhanjiang | 0 | 0 | -2 | 9 | [0,6] | [0, 6] |
| | Haikou | 20 | 21 | 16 | 7 | [14,36] | [14, 21] |
| $CO$ | Shenzhen | -36 | -40 | -36 | -39 | [-39, -38] | [-39, -38] |
| | Zhanjiang | -34 | -32 | -34 | -32 | [-62, -26] | [-34, -32] |
| | Zhuhai | -26 | -28 | -26 | -32 | [-32, -24] | [-32, -26] |
| | Sanya | -22 | -21 | -18 | -14 | [-20, -14] | [-20, -14] |
| | Haikou | -20 | -34 | -17 | -12 | [-15, -7] | [-15, -12] |
| | Guangzhou | -14 | -14 | -14 | -19 | [-29, -17] | [-19, -17] |
| $SO_2$ | Guangzhou | -44 | -46 | -46 | -47 | [-75, -50] | \* |
| | Zhuhai | -34 | -40 | -37 | -40 | [-59, -47] | \* |
| | Shenzhen | -22 | -24 | -23 | -32 | [-70, -18] | [-32, -23] |
| | Haikou | -19 | -18 | -16 | -20 | [-25, -20] | [-20, -20] |
| | Zhanjiang | -16 | -20 | -14 | -25 | [-22, -5] | [-22, -14] |
| | Sanya | 75 | 68 | 76 | 98 | [83, 94] | [83, 94] |

610