# Peer review of "Decoupling impacts of weather conditions on interannual variations in concentrations of criteria air pollutants in south China – constraining analysis uncertainties by using multiple analysis tools"

_Atmospheric Chemistry and Physics, 2022_

## Author Response (AR1)

November 7, 2022

Handling Editor, Dr. Rob MacKenzie

**Response Letter: Revision of Manuscript # acp-2022-502**

Dear Dr. Rob MacKenzie:

We have made revisions according to the comments. Here is a point-by-point summary of our response to comments and suggestions. The responses may be revised according to the final revision of the manuscript. The comments are listed first, and our responses follow each comment. We also checked and revised the whole manuscript and figures.

Best regards!

Sincerely, Xiaohong Yao, Ph.D. Ocean University of China

**Response to comments provided by Zongbo Shi**

We greatly appreciate Dr. Shi for providing the constructive comments, which have helped us improve the paper quality. We have addressed all of the comments carefully, as detailed below.

This paper used two machine learning techniques to remove the meteorological effects on air quality trend in South China. The paper contains new and publishable results. There are new developments in machine learning, which should be considered (see below). I suggest that the paper may be published after a major revision.

**Response:** In the revised manuscript we have included more parameters in machine learning modelling and also revised the abstract and re-organized Results and discussion accordingly.

**Line 19: define "in annual scale"**

**Response:** We meant "on an annual scale". One whole year data cover from May to the next April, e.g., the first-year covered from May 2014 to April 2015, and the last year covered from May 2020 to April 2021. We revised this part to: "To constrain the uncertainties in the calculated deweathered or decomposed hourly values, a self-developed method was applied to calculate the range of the deweathered percentage changes (DePCs) of air pollutant concentrations on an annual scale (each year covers May to the next April)."

**Line 21: define "consistency". Explain what does consistency of 70% or 30% means**

**Response:** In the revision, the part reads as "Consistent trends between the RFdeweathered and BRTs-deweathered concentrations and the ICEEMDAN-decomposed residuals of an air pollutant in a city were obtained in approximately 70% of a total of 42 cases (for seven pollutants in six cities), but consistent PCs calculated from the three methods, defined as standard deviation being smaller than 10% of the corresponding mean absolute value, were obtained in only approximately 30% of all the cases."

**Line 27: expand this section on results**

Abstract focused more on methods but not results. Is this a methodological development paper or usual academic paper? What is the take-home message? Abstract should be re-written.

**Response:** This study has several goals: (1) the first one is to assess the performance of several existing machine leaning methods used in decoupling weather impacts on concentrations of air pollutants; (2) the second one is to proposal the best approach that can be used for generating trends with minimum uncertainties; and (3) the third one is to extract trends for pollutants monitored in south China and validate the efficacy of

emission reduction policies. To balance these goals, we have added some trend analysis results in the abstract, which reads: "The calculated PCs from the deweathered concentrations and decomposed residuals were thus combined with the corresponding range of DePCs calculated from the self-developed method to gain the robust range of DePCs where applicable."

Methods: For secondary pollutants, it is important to include back trajectory clusters. Other met factors such as solar radiation might also be important. Please read relevant literature and include these important parameters.

**Response:** We agree with the reviewer on this point, and we have included more meteorological factors such as boundary layer height, total cloud cover, surface net solar radiation, surface pressure, and total precipitation, which are extracted from the European Centre for Medium Weather Forecasting's Reanalysis-5 (ERA5) hourly data (https://cds.climate.copernicus.eu/), and air mass clusters based on the Hybrid Single-Particle Lagrangian Integrated Trajectory (HYSPLIT) 72-hour back trajectories at an hourly resolution (https://www.ready.noaa.gov/HYSPLIT\_traj.php), and re-run the machine learning modeling accordingly. The newly generated results substantially increased the consistence between the observed and predicted PM2.5 concentrations with a slight increase in the consistency for O3, but no evident increase in consistency for other pollutants. The manuscript has been revised on basis of the newly generated results.

**Line 135: $R^2$ is not as good as other recent studies, why?**

**Response:**  $R^2$  values fluctuated due to different characteristic of various pollutants. For the newly generated results, the range of  $R^2$  in this study were 0.85~0.95 (PM2.5) and 0.88~0.95 (O3), comparable to those reported in earlier studies, e.g., 0.906±0.001 (PM2.5) and 0.63~0.92 (O3) (Hou et al., 2022; Ma et al., 2021).

Line 144: explain why using meteorological variables randomly resampled from the study period (2014-2020) is fit for purpose for this particular study? Note there are different methods - they are there for different purposes.

**Response:** We used the deputy design in RF and BRTs models for processing meteorological normalization. Based on invariant predicted hourly values during most of times in a year, the randomly resampled 1000 types of meteorological conditions from the study period (2014–2020) had demonstrated reasonably well representative. We also try 2000-time and 3000-time predications for meteorological normalization, the difference is negligible. The averaging 1000-time predictions has been also used in Hou et al (2022). The reference has been added.

Line 149: why not enhanced secondary pollution?

**Response:** Agree. The enhanced secondary pollution cannot be excluded and thereby added in the revision.

*Line 283: O3 changes are the result of emission changes of O3 precursors and changes in chemistry. Revise*

Response: Revised as suggested.

Line 301-302: I don't understand the argument here. Please explain in more detail

**Response:** The part has been revised as "Thus, the 39%-55% O3 increases from 2014 to 2020 likely attributed to the emission-driven enhanced O3 formation. In addition, the first three PCs values for (NO2+O3) were smaller than those of O3 by 10%-16%, which represented the reduced O3 depletion via the titration reaction (Li et al., 2019a; Wang et al., 2017)."

*Line 309: Would it be more reasonable to present*  $O_3$  *and*  $NO_2$ *, and then*  $O_3+NO_2$

Response: Agree. The order has been adjusted.

Figure 3: The prediction appears to be relatively poor. Fig. a shows three distinct areas. It appears to me There is something wrong -I would suggest that the authors check the codes and re-run the results, particularly including other parameters mentioned above.

**Response:** We checked and re-run the codes by adding other meteorological factors as above-mentioned, and adjusted the range of y-axis. The original y-axis didn't start from point (0,0) and were thereby corrected in the revision.

 $NO_2+O_3$  is often defined as Ox. But you cannot add these two together based on mass concentration. Please turn  $NO_2$  and  $O_3$  into ppb first and then add.

**Response:** The concentrations of  $(NO_2+O_3)$  were calculated by adding those of them with the molecular weight correction, i.e.,  $[NO_2+O_3] = [NO_2] *48/46+ [O_3]$ . This has been clarified in the revision. We are sorry for missing the information in the original method.

Discussions are inadequate, more or less repeating the results rather than an in-depth discussion. Two suggestions: interpret the results, in the contexts of literature and clean air policies; examine the implications of the results – e.g., what policies are effective and what are not. Suggest to remove all discussions in the Results, and move to Discussions as needed

**Response:** Discussion in the original manuscript aims to use the self-developed independent method to constrain the uncertainties of the percentage changes of air

pollutant annual average concentrations estimated by the two deweathered methods and one decomposed method. Based on the comments, the authors realize that it is misleading as an independent Discussion Section. In the revision, we re-organized Results and discussion Section. Discussion section in the original manuscript has been converted to Section 3.4 with subtitle as "Constraining analysis uncertainties". We hope that subtitle can help solve the misleading to some extent.

Air pollutant emissions have not been updated in the annual reports of ecology and environment issued by local governments at the city level since 2014 in China. In the south China, the air pollution has been largely relieved before 2014. The implemented clean air policies therein were not issued in public domain, except the national clean air policies, i.e., Air Pollution Prevention and Control Action Plan in China (2013-2017), and Three-year Action Plan to Fight Air Pollution (2019-2021). The national polices are too general to link with the concentration trends of air pollutants, particularly for no detailed implement measures and corresponding air pollutant emission data. Thus, the emission-driven trends in air pollutant concentrations are critical to accurately evaluate the achievement of every-three-year national targets in south China.

**Response to comments by Anonymous Referee #2**

We greatly appreciate this reviewer for providing the constructive comments, which have helped us improve the paper quality. We have addressed all of the comments carefully, as detailed below.

Analyzing long-term trends by excluding the effects of meteorological factors is critical in the assessment of anthropogenic air pollution factors. In this paper, the authors have used three different methods to decouple meteorological effects and investigate the trends of different pollutants in South China. I find the comparison of these three methods valuable and novel even though the trends were only consistent in 30% of the conditions between these approaches. The manuscript is well-written and has a proper flow to it. The problem statement and introduction are well-written. The discussion of results is clear. However, I think the method section should be expanded and better explained. Here are some general comments for improvement:

RF and BRTs Modeling can be explained better. In particular, how the train-test splitting was applied is not explained thoroughly as it is important in model development. Was this random or sequential? For time series with long-term trends, this split should not be applied randomly, as might be customary in most of the random forest models in other fields, and should be applied sequentially. This is due to the fact that random split will bring extra information to the test validation (e.g. seasonal or weekly trends) that should not be available to the test and cause data leakage.

**Response:** The software Packages used in this study designed the train-set splitting randomly. This has been clarified in the revision. In the revision, we also added "The independent input variables included temporal variables (hour, day, weekday, week and month), observational concentrations and meteorological parameters (ws, wd, at, rh, dp, blh, tcc, ssr, sp and tp), the top three most influential variables in each modeling were listed in Table S3." Since temporal variables (hour, day, weekday, week and month) have been added in the machine learning, the random or sequential train-test splitting should not affect the performance of machine learning prediction. Moreover, the authors agree with the software developers for the random train-set splitting by considering emission changes through the study period.

The modeling work needs a feature importance analysis. This is very important since some of the features might not add anything to the model and can be simply eliminated from the analysis. Also, it shows the most influential meteorological factor on the trends. Some additional information can be added to the discussion section about the reasons for observing some of the trends. For example, if authors hypothesize specific regulation(s) as the reason for a specific deweathered trend, that can be added in the discussion section in addition to the introduction.

Response: Thanks for the advice. The results have been added in Supporting

Information Table S3. However, no conclusive results can be obtained on the most influential meteorological factors regardless only top 1 and top 3 to be considered. Some of the results have been added in Results and discussion.

*The error or confidence intervals should be added to trend figures (e.g. figures 4 and 5).*

**Response:** The error bars of original annual averages have been added in the revision accordingly. Like all air quality modeling results, the predicted values have no error bars.

Line 37: change "..two-three year.." to "...two-three years" Line 128: change "predicated" to "predicted" Line 138: change "indicates" to "indicate" Line 193: change "decreases" to "decrease". Line 232: change "conducted to" to "conducted on" Line 239: change "obtained between" to "obtained by" Line 269: change "annul" to "annual"

Response: Thanks for the comments. We have revised the manuscript accordingly.

Line 100 and figure 1: "Hourly meteorological data … were obtained from the meteorological observational station at a nearby airport". The meteorology factors, especially wind direction, change rapidly spatially at nearshore sites similar to the ones used in this work. Please mention that the meteorological stations were the closest available to the air quality sites if that is the case. Otherwise, please try to use the closest possible station in your database. Also, this should be mentioned as a source of error in the analysis.

**Response:** In each city, the hourly averages of air pollutant concentrations at multiple sites in a city were used for machine learning. Thus, the input meteorological data should reflect synoptic weather conditions. Airports usually have a widely open space and the meteorological data at the nearest airports can be reasonably assumed as the synoptic weather conditions.

In the new version, it has been revised as "Like most of studies in the literature (Dai et al., 2021; Ma et al., 2021; Mallet, 2020; Vu et al., 2019; Wang et al., 2020), the meteorological data from the nearest airports were used for the two machine learning methods. The data reflected synoptic weather conditions and were particularly applicable for modelling the hourly averages of air pollutant concentrations at multiple sites in a city."

Figure 3: the range of predicted values is considerably smaller than the observed value. This is an inherited issue with RF and BRT models and should be explained in the text. **Response:** In the revision, we added "Note that two machine learning methods always underpredicted the larger values of  $PM_{2.5}$  concentrations which occurred less frequently. The same underprediction has also been reported in air quality modelling  $PM_{2.5}$  concentrations, which could be due to missing mechanisms enhancing formation of  $PM_{2.5}$  under poor dispersion conditions (Chang et al., 2020; Liu et al., 2021a; Shen et al., 2022; Zheng, et al., 2015). For these infrequent cases, the training for two machine learning methods may not be sufficient enough to yield good prediction."